

# rtpcr: a package for statistical analysis and graphical presentation of qPCR data in R

Ghader Mirzaghaderi

Plant Production and Genetics, University of Kurdistan, Sanandaj, Iran

## ABSTRACT

**Background**. Quantitative real-time polymerase chain reaction (qRT-PCR or qPCR) is widely used in molecular biology research. Various analysis methods are employed to interpret qPCR data and measure mRNA levels of target genes under different experimental conditions.

**Results**. The rtpcr package was developed for amplification efficiency calculation, statistical analysis, and graphical presentation of qPCR data in R. It uses a general calculation methodology that accommodates up to two reference genes and amplification efficiency values, including the Pfaffl method. Depending on the experimental design, rtpcr functions apply a $t$-test (for experiments with a two-level factor), analysis of variance (ANOVA), or analysis of covariance (ANCOVA) (for experiments with more than two levels or factors) to calculate fold change (FC) or relative expression (RE) of a target gene. The functions also provide standard errors and confidence intervals for the means and support statistical mean comparisons. To facilitate usage, the package includes example datasets. It also offers ggplot-based visualizations with customizable arguments, allowing users to tailor the graphical output.

**Conclusions**. In summary, the rtpcr package is a useful and user-friendly tool for analyzing real-time PCR data from experiments involving up to three different factors. Built on a general methodology, it provides robust calculations and comprehensive graphical outputs, making it a valuable resource for researchers working with qPCR data.

## BACKGROUND

Quantitative real-time polymerase chain reaction (qRT-PCR also known as qPCR), is a powerful analytical tool that is able to quantify nucleic acids using reference sequences. The technique's sensitivity, specificity, and broad quantification range make it the gold standard for the detection and quantification of DNA and RNA sequences. Nowadays, relative quantification using qPCR is primarily used in the field of molecular genetics, genomics and functional transcriptomics to perform gene expression analysis in biological experiments (*Harshitha & Arunraj, 2021*).

Corresponding author
Ghader Mirzaghaderi,
mirzaghaderi@gmail.com

qPCR utilizes the increased fluorescence from a reporter molecule as the template is exponentially amplified. SYBR Green is a commonly used chemical that binds to double-stranded DNA during the primer extension step of the polymerase reaction (*Boulter et al., 2016*). Specialized oligonucleotide probes like TaqMan hybridization probes are also used as fluorescent reporters for targeted RT-qPCR assays (*Navarro et al., 2015*).

Reliability of the qPCR results depends on the application of robust mathematical methodologies that ensure accurate data analysis and outputs. Various mathematical approaches have been developed for qPCR data analysis (*Pabinger et al., 2014*; *Ritz & Spiess, 2008*). The basic method for expression estimation of a gene between conditions relies on the calculation of fold changes (FC) using the PCR amplification efficiency (E) and the threshold cycle (syn. crossing point or $C_T$). Two basic mathematical methods are commonly used based on FC namely the Livak method and the Pfaffl method. The Livak approach (*Livak & Schmittgen, 2001*), also known as the $2^{-\Delta\Delta CT}$ method, stands out for its simplicity and widespread use where the fold change (FC) expression of a target gene ($2^{-\Delta\Delta CT}$) in Treatment condition (Tr) compared to Control condition (Co) is calculated according to the following Eq. (1):

$$FC = \frac{2^{-(C_{T_{target}}-C_{T_{ref}})_{Tr}}}{2^{-(C_{T_{target}}-C_{T_{ref}})_{Co}}} \tag{1}$$

$$= 2^{-[(C_{T_{target}}-C_{T_{ref}})_{Tr}-(C_{T_{target}}-C_{T_{ref}})_{Co}]}$$

$$= 2^{-(\Delta CT_{Tr}-\Delta CT_{Co})}.$$

Here, $\Delta C_T$ is the difference between two $C_T$ values (*e.g.*, $CT_{target}-CT_{ref}$) in treatment or control condition, and target and ref are target gene and reference genes, respectively. This method assumes that both the target and reference genes are amplified with efficiencies close to 100%, allowing for the relative quantification of gene expression levels (*Livak & Schmittgen, 2001*).

On the other hand, the Pfaffl method (*Pfaffl, Horgan & Dempfle, 2002*) offers a more flexible approach by accounting for differences in amplification efficiencies between the target and the reference genes. This method adjusts the calculated expression ratio by incorporating the specific amplification efficiencies, thus provides a more accurate representation of the relative gene expression levels (*Pfaffl, Horgan & Dempfle, 2002*). The Pfaffl formula can be written as Eq. (2):

$$FC = \frac{E^{-(C_{T_{Tr}}-C_{T_{Co}})_{target}}}{E^{-(C_{T_{Tr}}-C_{T_{Co}})_{ref}}}. \tag{2}$$

Many statistical tools and analysis codes have been developed for the statistical analysis of qPCR data in a stand alone format (*Yuan et al., 2006*), for the R platform (*Ahmed & Kim, 2018*; *Li et al., 2022*; *Ritz & Spiess, 2008*) or SAS software (*Yuan et al., 2006*). The rtpcr package is a comprehensive tool designed for the analysis of qRT-PCR data in R, providing a set of functions that allow researchers to perform various analyses on their qRT-PCR data using the cycle threshold ($C_T$) and efficiency values (E) under different experimental conditions.

## IMPLEMENTATION

The rtpcr package was developed for the R environment (http://www.r-project.org) in the major operating systems and its source code, binary format, and under-development version are freely available at CRAN (https://cran.r-project.org/web/packages/rtpcr/index. html) and Github (https://github.com/mirzaghaderi/rtpcr). The package functions are mainly based on the calculation of efficiency-weighted $\Delta C_T$ (w$\Delta C_T$) values from target and reference gene $C_T$ (Eq. (3)). w$\Delta C_T$ values are weighted for the amplification efficiencies as below. Portions of this text were previously published as part of a preprint (*Mirzaghaderi, 2024*):

$$w\Delta C_T = \log2(E_{target}).CT_{target} - \log2(E_{ref}).CT_{ref}. \tag{3}$$

From the arithmetic mean $w\Delta C_T$ values over biological replicates, relative expression (RE, $\Delta C_T$ method) of a target gene can be calculated for each condition according to Eq. (4):

$$RE = 2^{-\overline{w\Delta CT}}. \tag{4}$$

The rtpcr package considers efficiency values, thus the results match the Pfaffl method. If all input efficiency values are 2, $2^{-\Delta\Delta C_T}$-based results are returned. The average fold change (FC, $\Delta\Delta C_T$ method) expression statistics and graphs are returned for the target gene based on Eq. (5):

$$FC = 2^{-(\overline{w\Delta CT}_{Tr} - \overline{w\Delta CT}_{Co})}. \tag{5}$$

Because both the relative expression and fold change expression follow a lognormal distribution (*Derveaux, Vandesompele & Hellemans, 2010*; *McDavid et al., 2012*), a normal distribution is expected for the w$\Delta C_T$ or w$\Delta\Delta C_T$ values making it possible to apply t-tests or analysis of variance to them. w$\Delta C_T$ values can be statistically compared and standard deviations and confidence intervals are calculated, but the transformation $y = 2^{-x}$ is applied in the final step to report the results. Here, a brief methodology is presented but details about the mathematical calculations and statistical analyses are available in *Ganger, Dietz & Ewing (2017)* and *Ganger et al. (2020)*. In the rtpcr package, model creation and analysis of variance is done using the lmer (*Bates et al., 2015*) function of the lmerTest (*Kuznetsova, Brockhoff & Christensen, 2017*) package. lmerTest::lmer fits a linear mixed model and provides *p*-values for fixed effects in the analysis of variance (ANOVA) and summary output. For this, the biological replicate is served as a random effect. Means are compared using the emmeans function (*Searle, Speed & Milliken, 1980*) from the model. The standard error (standard deviation/sqrt(n)) for average FC or RE is calculated in different ways in literature. This statistic is calculated based on *Taylor et al. (2019)* by the rtpcr package according to Eq. (6) for average FC. For the standard error of the RE means, RE is used instead of FC.

$$Lower.se = 2^{\log2(FC) - se_{w\Delta CT}} \tag{6}$$

$$Upper.se = 2^{\log2(FC) + se_{w\Delta CT}}.$$

# RESULTS

The rtpcr package was developed for amplification efficiency calculation, statistical analysis, and graphical representation of qPCR data in R. It accepts up to two reference genes and amplification efficiency values. Based on the experimental conditions, the functions of the rtpcr package use a $t$-test, analysis of variance, or covariance (for cases with more than two factors) to calculate the fold change (FC) or relative expression (RE). The functions provide standard errors and confidence intervals for FC or RE means and apply statistical mean comparisons. Different sample data sets were added to the rtpcr package and included in the examples to facilitate the usage of the package functions. The rtpcr package further provides editable ggplots with various editing arguments. Some functionalities of the rtpcr package are presented in Fig. 1 and a more detailed representation of the rtpcr application is shown in Fig. S1.

## Input data frame structure

The input data set should be prepared as shown in example data sets of the rtpcr package. The column structure and arrangement of the data frames should follow the format indicated in Tables 1 and 2, as shown in the package help and the vignettes (https://cran.r-project.org/web/packages/rtpcr/vignettes/vignette.html). To see example data sets included in the rtpcr package, you can run the following command in R: data(package = "rtpcr"). This will display a list of the available example data sets within the rtpcr package. You can then load a specific data set by running its name. Except for the $t$-test analysis which requires a specific data structure, factor columns should appear first in the data frame followed by blocking factor (if available), biological replicates, target gene efficiencies, $C_T$ values of the target gene, the reference gene efficiencies, and the $C_T$ values of the reference gene, respectively. The data frame can contain one to three factor columns, depending on the experimental design.

This recommended column structure ensures compatibility with the various analysis functions provided by the rtpcr package, such as the analysis of variance and analysis of covariance methods, which require the data to be organized in this specific way. For the $t$-test analysis, the data structure may differ slightly, as it needs to be in a format that allows for the comparison of two experimental conditions. The package documentation and examples will provide guidance on the appropriate data structure for this specific analysis.

## Amplification efficiency

The rtpcr package in R provides a comprehensive set of functions for the analysis of real-time PCR (qPCR) data. A brief explanation of the key functions is presented in Fig. S1. An important function in the rtpcr package is the 'efficiency' function. This function is used to calculate the amplification efficiency of genes based on the provided data. The input data frame for the 'efficiency' function should have a specific column structure, with the first column containing the dilution information, followed by the $C_T$ value columns for each target gene. The 'efficiency' function takes this input data and calculates the amplification efficiency of the target genes. It presents the related standard curves, along with the slope, efficiency, and $R^2$ statistics (Fig. 1A). Additionally, the function performs

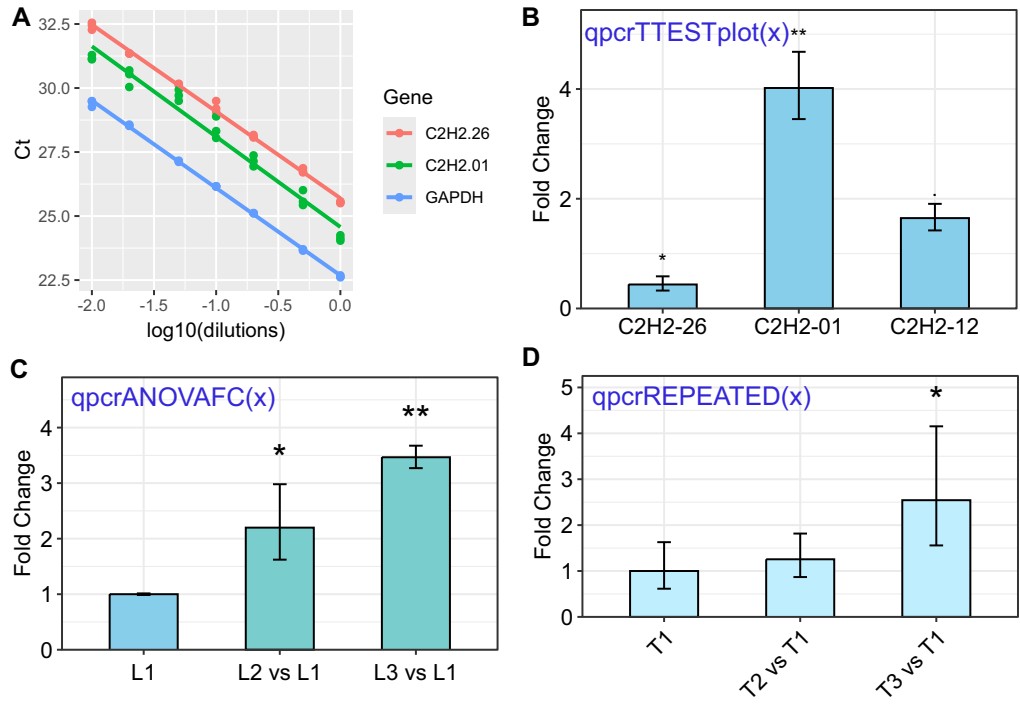

**Figure 1** **Some of the rtpcr package functionalities for the analysis of the qPCR data.** The used functions are presented in blue. (A) Standard curve and the amplification efficiency analysis of three genes using the 'efficiency(x)' function. (B) Average fold changes of three target genes relative to the control condition computed by unpaired $t$-tests *via* the 'qpcrTTESTplot' function. (C) Plot of average Fold changes produced by the 'qpcrANOVAFC' function where the level 1 condition (L1) has been selected as calibrator. (D) Fold change expression graph of a target gene produced by 'qpcrREPEATED' function. Error bars can be set as standard deviation or confidence interval.

statistical pairwise comparisons of the slopes to determine if the amplification efficiencies of genes are significantly different.

## Fold Change (FC) analysis: t-test

The expression of a target gene under two different conditions, such as control and treatment, can be presented as the average fold change of the target gene in the treatment condition relative to the control condition. This analysis is performed by the 'qpcrTTEST' and 'qpcrTTESTplot' functions, which apply a $t$-test to any number of genes that have been evaluated under control and treatment conditions. The analysis can be done for both unpaired or paired samples. A p.adj argument has been added to qpcrTTEST and qpcrTTESTplot functions to adjust the pvalues, in which the default adjustment method is set to "BH". An example of the resulting output of the 'qpcrTTEST' function is shown in Table 3, which contains the target gene names, fold changes, confidence limits, $p$-values, standard errors (se), and lower and upper standard errors. The 'qpcrTTEST' function includes the 'var.equal' argument, which, if set to 'FALSE', performs the $t$-test under the unequal variances hypothesis. The 'qpcrTTESTplot' function, which performs the $t$-test and displays the results in a bar plot, automatically adds appropriate significance

**Table 1  Column arrangement of the input data frame for use in the rtpcr package.** To see example data sets, in the rtpcr package rundata (package = "rtpcr"). Example data sets can be presented by running the name of each data set. targetE and refE: amplification efficiency columns for target and reference genes, respectively. targetCt and refCt: Ct columns for target and reference genes, respectively.

| Analysis type | Column arrangement of the input data set | Name of the example data sets |
|---|---|---|
| Amplification efficiency | Dilutions - targetCt - refCt | data_efficiency |
| $t$-test (accepts multiple genes) | condition (control level first) - gene (ref gene(s) last) - efficiency - Ct | data_ttest |
| ANOVA or ANCOVA (Up to three factors) | factor1 - rep - targetE - targetCt - refE - refCt | data_1factor |
| | factor1 - factor2 - rep - targetE - targetCt - refE - refCt | data_2factor |
| | factor1 - factor2 - factor3 - rep - targetE - targetCt - refE - refCt | data_3factor |
| ANOVA or ANCOVA with blocking | factor1 - block - rep - targetE - targetCt - refE - refCt | |
| | factor1 - factor2 - block - rep - targetE - targetCt - refE - refCt | data_2factorBlock |
| | factor1 - factor2 - factor3 - block - rep - targetE - targetCt - refE - refCt | |
| With two reference genes | . . . . . . rep - targetE - targetCt - ref1E - ref1Ct - ref2E - ref2Ct | |
| Calculating biological replicated | . . . . . . biologicalRep - techcicalRep - Etarget - targetCt - Eref - refCt | data_withTechRep |
| | . . . . . . biolRep - techRep - Etarget - targetCt - ref1E - ref1Ct - ref2E - ref2Ct | |

**Notes.**
For ANOVA and ANCOVA analysis, each line in the input data set belongs to a separate individual (reflecting a non-repeated measure experiment).

**Table 2  Repeated measure data structure and column arrangement required for the 'qpcrREPEATED' function.** targetE and refE: amplification efficiency columns for target and reference genes, respectively. targetCt and refCt: Ct columns for target and reference genes, respectively. In the "id" column, a unique number is assigned to each individual, *e.g.* all the three number 1 indicate a single individual.

| Column arrangement of the input data set | Name of the example data sets |
|---|---|
| id - time - targetE - targetCt - ref1E - ref1Ct | data_repeated_measure_1 |
| id - time - targetE - targetCt - ref1E - ref1Ct - ref2E - ref2Ct | |
| id - treatment - time - targetE - targetCt - ref1E - ref1Ct | data_repeated_measure_2 |
| id - treatment - time - targetE - targetCt - ref1E - ref1Ct - ref2E - ref2Ct | |

**Notes.**
To see example data sets, in the rtpcr package run data (package = "rtpcr"). Example data sets can be presented by running the name of each data set.

signs ('**', '*', or '.') on top of the bars based on the $t$-test $p$-values (Fig. 1B). Similarly, in single- or multi-factorial experiments, fold change (FC) analysis can be performed for each of the factors using the 'qpcrANOVAFC' function. One of the factor levels can be selected as the reference level, allowing for the comparison of gene expression across different experimental conditions. These functions in the rtpcr package provide a comprehensive set of tools for researchers to analyze and interpret gene expression

**Table 3  The fold change (FC) analysis.** An example output table of fold change (FC) analysis of three genes evaluated under control and treatment conditions. The output was produced by the 'qpcrTTEST' function of the rtpcr package.

| Gene | FC | LCL | UCL | *p* value | se | Lower.se | Upper.se |
|------|------|------|------|------|------|------|------|
| C2H2-26 | 0.4373 | 0.1926 | 0.9927 | 0.0488 | 0.4218 | 0.3264 | 0.5858 |
| C2H2-01 | 4.0185 | 2.4598 | 6.5649 | 0.0014 | 0.2193 | 3.4518 | 4.6782 |
| C2H2-12 | 1.6472 | 0.9595 | 2.8279 | 0.0624 | 0.2113 | 1.4228 | 1.907 |

**Notes.**

FC, The average fold change of the target gene in the treatment condition relative to the control condition; LCL, lower confidence limit of the FC; UCL, upper confidence limit; se, standard error of the fold change; Lower.se, lower standard error; Upper.se, upper standard error.

data from qPCR experiments, enabling them to draw meaningful conclusions about the differential expression of target genes under various experimental conditions.

## Fold change (FC) analysis: analysis of variance

If there is one factor with more than two levels or more than one factor in the experiment, the qpcrANOVAFC function can be used for the fold change analysis of the levels of each factor. A partial output of this function has been presented in Table 4. The qpcrANOVAFC function applies both ANOVA and analysis of covariance (ANCOVA) analysis to the data of a single-factor or a multi-factorial experiment. If there are multiple factors, the fold change (FC) value calculations for the mainFactor.column and the statistical analysis are performed based on a full model factorial experiment by default. However, if ANCOVA is defined for the analysisType argument, the FC values are calculated for the levels of the mainFactor.column, and the other factors are used as covariate(s) in the analysis. It is important to consider the output analysis of variance table, as if an interaction between the main factor and another factor or covariate is significant, statistical comparisons of FC values may not be appropriate between the levels of a factor alone. ANCOVA is used when a factor is affected by uncontrolled quantitative covariate(s). For example, suppose that the $\Delta C_T$ of a target gene in a plant is affected by temperature. The gene may also be affected by drought. Since we already know that temperature affects the target gene, we are interested in understanding if the gene expression is also altered by the drought levels. We can design an experiment to study the gene behavior at both temperature and drought levels simultaneously. The drought is another factor (the covariate) that may affect the expression of our gene under the levels of the first factor, *i.e.,* temperature. The data of such an experiment can be analyzed using ANCOVA or ANOVA based on a factorial experiment using the qpcrANOVAFC function. The qpcrANOVAFC function performs FC analysis even if there is only one factor (without a covariate or factor variable). It also returns a bar plot of the FC values along with the standard errors.

The qpcrMeans function also performs FC analysis using a model produced by the qpcrANOVAFC or qpcrREPEATED functions. As an advantage, qpcrMeans function returns all the pairwise statistical comparisons for any effects in the model, including simple effects, interactions, and slicing from the ANOVA models. However, ANCOVA models returned by the rtpcr package only include simple effects.

**Table 4** **The output table of fold change analysis.** An example output table of fold change analysis of one gene evaluated under three different levels of a factor. The output was produced by the 'qpcrANOVAFC' function of the rtpcr package applied on the 'data_1factor' sample data. 'L1' has been selected as the check or reference level.

| Contrast | FC | *p* value | sig | LCL | UCL | se | Lower.se | Upper.se |
|---|---|---|---|---|---|---|---|---|
| L1 | 1.0000 | 1.0000 | | 0.0000 | 0.0000 | 0.0208 | 0.9857 | 1.0145 |
| L2 *vs* L1 | 2.1987 | 0.0285 | * | 0.9514 | 5.0812 | 0.4388 | 1.6221 | 2.9803 |
| L3 *vs* L1 | 3.4661 | 0.0061 | ** | 1.4999 | 8.0101 | 0.0841 | 3.2698 | 3.6742 |

**Notes.**

FC, The average fold change of the target gene in the treatment condition compared to the control condition; sig, * and ** shows significant FC expression at 0.05 and 0.01 significance levels, respectively; LCL, lower confidence limit of the FC; UCL, upper confidence limit; se, standard error of the FC; Lower.se, lower standard error; Upper.se, upper standard error.

**Table 5** **The output table of relative expression.** An example output table of relative expression of a target gene evaluated under a two factorial experiment. The output produced by the 'qpcrANOVARE' function of the rtpcr package.

| Factor1 | Factor2 | RE | LCL | UCL | se | Lower.se | Upper.se | Letters |
|---|---|---|---|---|---|---|---|---|
| S | 0.5 | 2.9545 | 2.047 | 4.2644 | 0.0551 | 2.8438 | 3.0695 | a |
| R | 0.5 | 0.9837 | 0.6815 | 1.4198 | 0.0841 | 0.928 | 1.0427 | b |
| S | 0 | 0.7916 | 0.5485 | 1.1426 | 0.2128 | 0.683 | 0.9174 | b |
| R | 0.25 | 0.624 | 0.4323 | 0.9006 | 0.4388 | 0.4604 | 0.8458 | bc |
| S | 0.25 | 0.4126 | 0.2859 | 0.5956 | 0.254 | 0.346 | 0.492 | cd |

**Notes.**

RE, Relative expression of the target gene relative to the reference gene in each specific condition; LCL, lower confidence limit; UCL, upper confidence limit; se, standard error; Lower.se, lower standard error; Upper.se, upper standard error; Letters, Means that does not share a letter in common, have significant difference.

## Relative expression (RE) analysis

The 'qpcrANOVARE' function can perform ANOVA analysis of the relative gene expression ($\Delta$Ct method) for one- to three-factor experiments. The package generates relative statistics and 'ggplot2'-derived graphs (*Wilkinson, 2016*). If available, the blocking factor can also be handled. The output of the 'qpcrANOVARE' function includes a table with relative expression values, grouping letters, standard deviations, and post-hoc testing of means along with the significance and confidence interval (Table 5). The standard deviation for each mean is derived from the back-transformed raw wDCt values of biological replicates.

An outstanding feature of the rtpcr package is providing publication-ready bar plots with various controlling arguments for a lot of graphical aspects of plots. A sample of the output plots are presented in Fig. 1 and Fig. S1. The RE table from the 'qpcrANOVARE' function can be used by plot functions to generate a bar plot. 'qpcrTTESTplot' and 'qpcrANOVAFC' functions also generate fold change plots directly from the raw data. The bar plots produced by the rtpcr package can further be edited by the ggplot2 functions.

## Checking residuals

If the residuals from a *t*-test or a linear model (lm) object do not show homogeneity of variances or are not normally distributed, the assumptions for the *t*-test and ANOVA analyses may be violated. In such cases, the statistical results might not be reliable. The qpcrTTEST and qpcrTTESTplot functions in the rtpcr package include the var.equal

argument. When set to FALSE, these functions perform the $t$-test under the unequal variance hypothesis, by which can address the issue of heterogeneous variances. However, in other cases, such as ANOVA analysis, it may be more appropriate to apply non-parametric tests instead of the standard ANOVA. Currently, non-parametric tests have not been built in the rtpcr package but some options include the Mann–Whitney test (a non-parametric alternative to the $t$-test that can be used to test the difference between the medians of two populations using independent samples) and the Kruskal-Wallis test (a non-parametric alternative to one-way ANOVA can be used to test the difference between the medians of multiple populations). These non-parametric tests do not rely on the assumptions of normality and homogeneity of variances, making them more robust when these assumptions are violated.

### Mean of the technical replicates

Calculating the mean of technical replicates and getting an output table appropriate for subsequent analysis in rtpcr can be done using the 'meanTech' function. For this, the input data set should follow the column arrangement of the 'data_withTechRep' example data set. The grouping columns needs to be specified using the 'groups' argument of the 'meanTech' function.

## DISCUSSION

rtpcr is an open-source R package that covers a lot of aspects of qPCR data analysis including efficiency and fold change analysis, statistical comparisons and graph production from single and multi-factorial experiments with or without a blocking factor. It accepts one or two reference genes, performs post-hoc testing, and provides standard error and confidence limits.

Different R packages have been developed for the analysis of qPCR data with different capabilities (*Ahmed & Kim, 2018*; *Pabinger et al., 2014*; *Yuan et al., 2006*). For example, chipPCR (*Rödiger, Burdukiewicz & Schierack, 2015*) can handle high throughput qPCR data and qpcR performs sigmoidal model selection for the analysis of the real-time PCR data (*Ritz & Spiess, 2008*). Some packages such as chipPCR, qpcR and FPK-PCR (*Lievens et al., 2012*) calculate the $C_T$ values from raw florescence data and some such as ddCT (*Zhang, Ruschhaupt & Biczok, 2013*), dpcR, EasyqpcR, HTqPCR (*Dvinge & Bertone, 2009*), NormqPCR (*Perkins et al., 2012*), qpcrNorm, pcr (*Ahmed & Kim, 2018*) and qPCRtools (*Li et al., 2022*) require $C_T$ values for the analysis. These packages also differ in type of quantification and analysis, amplification efficiency handling, *post-hoc* comparison, error calculation, and graph presentation (Table 6). The main advantages of the rtpcr package over the other R packages are performing expression analysis based on both $\Delta C_T$ and $\Delta\Delta C_T$ methods for different experimental conditions and accounting for given amplification efficiencies matching the Pfaffl method. Using the rtpcr package, it is also possible to account for efficiency value for each primer or each reaction which has been recommended to get a more precise quantification result (*Ruijter et al., 2021*; *Ruiz-Villalba, Ruijter & van den Hoff, 2021*). The provided sample data frames for all the analysis types and a documentation enables all R users with minimum knowledge of data handling to

Table 6 **rtpcr package functionality.** Comparing functionalities of the rtpcr package with available R packages for qPCR data analysis.

| Package name | Efficiency calculation | efficiency values[a] | Fold change expression | Error calc. | Norm. | NA handling | Graphs | Stats. | Experimental designs[b] | MIQE[c] |
|---|---|---|---|---|---|---|---|---|---|---|
| chipPCR | + | | | | | + | + | | | + |
| ddCT | | + | + | | + | | + | + | | + |
| dpcR | | | | | + | | + | + | | + |
| EasyqpcR | + | + | + | | + | + | | | | + |
| HTqPCR | | | + | | + | + | + | + | + | |
| NormqPCR | | + | + | | + | + | | | | + |
| qpcR | + | + | + | + | + | + | + | | | + |
| qpcrNorm | | | | | + | | + | + | | |
| pcr | + | | | + | + | | + | + | | + |
| rtpcr | + | + | + | + | + | | + | + | + | + |

**Notes.**

[a]Can include PCR amplification efficiency values for each replicate in fold change or relative expression analysis.

[b]Data from multi-factorial and repeated measure designs can be handled.

[c]Follows MIQE (Minimum Information for Publication of Quantitative Real-Time PCR Experiments) approvals for reporting RT-qPCR results.

simply use the rtpcr package. In conclusion, the rtpcr was built based on a general method with both calculational and graphical output capabilities for analyzing qPCR $C_T$ values from the experiments with up to three different factors.

### Funding
The author received no funding for this work.

### Competing Interests
The author declares that he has no competing interests.

### Author Contributions
- Ghader Mirzaghaderi conceived and designed the experiments, performed the experiments, analyzed the data, prepared figures and/or tables, authored or reviewed drafts of the article, and approved the final draft.

### Data Availability
The rtpcr package source is available at CRAN:

- https://doi.org/10.32614/CRAN.package.rtpcr.

### Supplemental Information
Supplemental information for this article can be found online at http://dx.doi.org/10.7717/peerj.20185#supplemental-information.

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
