# Peer review of "rtpcr: a package for statistical analysis and graphical presentation of qPCR data in R"

_PeerJ, doi:10.7717/peerj.20185_

## Round 0.1 · original submission · Major Revisions

Reviewer 1 ·

Basic reporting

Background: “Biology” not “Bioligy”; Line 45: “Reliability”; Line 57: “… developed for qPCR analysis.”; Line 60: “… based on FC, namely the Livak and Pfaffl method”; in general: don’t say “of the qPCR data” but “of qPCR data”; line 86: … of qRT-PCR data in R, providing a set… “; line 107: “considers” not “concideres”; line 109: “…, expression statistics and graphs are…”; 116: “t-tests”; line 118: remove “used”; 119: “analyses”; line 137: “. The rtpcr package…”; line 139: “… a more detailed…”; line 149: “… should appear first…”; The sentences in lines 263+264 and 269+270 are redundant. Remove the first one and put the citations in the second;

Experimental design

Line 101: Please state which kind of mean is employed: geometric or arithmetic?
Lines 120+121: Please state why lmer and lmerTest functions are employed, as both pertain to a linear mixed-effects analysis. Why was this algorithm selected instead of standard R lm or anova functions? In the lmer model, what is defined as fixed and random effects? Please be more detailed and verbose on the selection of this approach.
Formula 6, lines 126+127: Please define all variables in the text, especially the se needs to be defined as the standard error of the mean, sigma/sqrt(n).
Lines 176ff and Figure 1B: It is not really clear, if the p-values from multiple t-tests on the same data set are corrected for multiple testing, i.e. Bonferroni. This is vital, especially when a large number of genes are pairwise compared.
Lines 271ff: The author states, and it is well-known, that a plethora of R-packages for qPCR analysis exist. It would be of great benefit to the dedicated readership if the author could include a table that provides a rough overview on how all these packages differ w.r.t. efficiency estimation, delta-Ct calculation, FC estimation and plotting capabilities, and especially, where the rtpcr package fits in and where it offers additional functionality.

Validity of the findings

All underlying data have been provided; they are robust, statistically sound, & controlled.

Additional comments

In this work, Mirzaghaderi describes the development of the R package rtpcr, available on CRAN, that conducts the calculation of qPCR efficiencies, efficiency-weighted delta-Cq values, statistical analysis (t-test, ANOVA, ANCOVA) of differential gene expression and graphical display thereof. Although this workflow would be possible with a combination of several packages (for instance qpcR) and base R functions for statistical analysis and barplot/lineplots, the package might facilitate this kind of analysis for the not so R-savvy scientist. In principle, this is a nice paper and package, however I have a few remarks that need to be addressed to render this paper publishable (Minor Revision).

Reviewer 2 ·

Basic reporting

The author introduced an R package for the analysis and visualization of qPCR data. The article is written in formal English that flows. Sections are well structured, with self-contained figures validating the claims made in the texts.

The manuscript falls short in one major aspect. Since this is a method/tool paper, the author should explain why having the addition of this R package can benefit the community and build upon existing toolkits. To do so, the author should compare this R package to other tools, such as the R packages that the authors cited (4,9,10) as well as non-R platforms, in the Result section. In particular, how does this R package excel at the visualization task when many users are using Prism nowadays, which can generate standardized plots without any scripting requirements.

A minor issue is that the entire paragraph on Line 48-56 in the Background sections seems out of place. How is this information related to the purpose of this article?

Experimental design

The author should implement non-parametric statistical tests in this R package. Non-parametric methods for either two or multiple groups has gained much more popularity today as many more researchers are starting to realize that their data do not follow a normal distribution, especially in biological studies where the sample size is usually quite small. Since the author already mentioned non-parametric tests in Line 239-252, it should be added to the package.

Validity of the findings

As mentioned before, the authors should state how this R package adds values, whether positive or not, to the current literatures. While the authors briefly discusses that other R packages does different things, the actual differences between them and this package was not highlighted.

---

## Round 0.2 · accepted · Accept

Dear Author,

Your paper has been revised. It has been accepted for publication in PeerJ Computer Science. Thank you for your fine contribution.

Reviewer 1 ·

Basic reporting

All issues have been removed adequately, and the manuscript is now ready for publication.

Experimental design

-

Validity of the findings

-

Reviewer 2 ·

Basic reporting

The author addressed all comments from the previous round of review.

Experimental design

-

Validity of the findings

-